# Diagnostic Performance and Tolerability of Saliva and Nasopharyngeal Swab Specimens in the Detection of SARS-CoV-2 by RT-PCR

Jaakko Ahti,[a,b] Riikka Österback,[c,d] Anniina Keskitalo,[c,d] Kati Mokkala,[e] Siina Vidbäck,[e] Ville Veikkolainen,[e] Tytti Vuorinen,[c,d] Ville Peltola,[a,b] Antti J. Hakanen,[c,d] Matti Waris,[c,d] Miia Laine[c,d]

aDepartment of Paediatrics and Adolescent Medicine, University of Turku, Turku, Finland

bDepartment of Paediatrics and Adolescent Medicine, Turku University Hospital, Turku, Finland

cDepartment of Clinical Microbiology, Turku University Hospital, Turku, Finland

dInstitute of Biomedicine, University of Turku, Turku, Finland

eWallac Oy, PerkinElmer, Turku, Finland

**ABSTRACT** Saliva is a promising alternative for a nasopharyngeal swab (NPS) in specimen collection to detect SARS-CoV-2. We compared the diagnostic performance and tolerability of saliva collection versus NPS in a clinical setting. Paired NPS and saliva specimens were collected sequentially from participants ($n = 250$) at the Turku University Hospital drive-in coronavirus testing station in the spring of 2022, with Omicron BA.2 as the dominant SARS-CoV-2 variant. Discomfort and preference for the sampling method were assessed. The specimens were analyzed for SARS-CoV-2 using real-time multiplex reverse transcriptase PCR (RT-PCR) with a laboratory-developed test (LDT) and two commercial kits (PerkinElmer SARS-CoV-2 and PerkinElmer SARS-CoV-2 Plus) for several target genes. Among the 250 participants, 246 had respiratory symptoms. With LDT, SARS-CoV-2 was detected in 135 and 134 participants from NPS and saliva, respectively. Of the 250 specimens, 11 gave a discordant outcome, resulting in excellent agreement between the specimen types (Cohen's kappa coefficient of 0.911; $P = 0.763$). The cycle threshold ($C_T$) values of LDT and commercial kit target genes were significantly lower from NPS than from saliva. A total of 172 (69%) participants assessed saliva sampling as more tolerable than NPS ($P < 0.0001$). Our findings present saliva as an applicable alternative for SARS-CoV-2 diagnostics. However, the lower $C_T$ values obtained from NPS indicate that NPS may be a slightly more sensitive specimen type. Participants preferred saliva sampling, although delivering an adequate volume of saliva was challenging for some participants.

**IMPORTANCE** The extensive testing of SARS-CoV-2 is vital in controlling the spread of COVID-19. The reference standard for specimen collection is a nasopharyngeal swab (NPS). However, the discomfort of NPS sampling, the risk of nosocomial infections, and global material shortages have accelerated the development of alternative testing methods. Our study demonstrates that patients tolerate saliva sampling better than NPS. Of importance, although the RT-PCR qualitative test results seem to correspond between NPS and saliva, we show significantly lower $C_T$ values for NPS, compared to saliva, thus contradicting the suggested superiority of the saliva specimen over NPS in the detection of the Omicron variants of SARS-CoV-2. Future research is still required to enable individual planning for specimen collection and to determine the effects of different SARS-CoV-2 variants on the sensitivity of the saliva matrix.

**KEYWORDS** COVID-19, nasopharyngeal swab, PCR, SARS-CoV-2, saliva, tolerability

Address correspondence to Miia Laine, miia.laine@tyks.fi.

The authors declare a conflict of interest. Kati Mokkala, Siina Vidbäck, and Ville Veikkolainen are PerkinElmer employees. The institution (Turku University Hospital) received financial support from Wallac Oy, PerkinElmer, to perform the study. The funder had no role in the data interpretation, or the decision to submit the work for publication.

The widescale diagnostic testing of severe acute respiratory syndrome coronavirus 2 (SARS-CoV-2) has been a crucial method in controlling the spread of COVID-19 (1, 2). At present, the reference standard for the diagnostic testing of SARS-CoV-2 is a reverse transcriptase PCR (RT-PCR) test that is performed on a specimen collected from the nasopharynx (2). Although nasopharyngeal swab (NPS) sampling is the recommended method for diagnostic specimen collection, there are some challenges in the sample collection procedure, and these challenges highlight the importance of the development of an alternative testing method (3, 4).

The complications of NPS collection are rare and are often associated with predisposing conditions, such as previous rhino surgery, septal deviation, and coagulopathy (5, 6). These complications range from mild epistaxis to potentially fatal cerebrospinal fluid leakage (5–8). While nearly all of the examinations succeed without complications, most of the patients find the testing to be at least uncomfortable or even painful (9, 10). This might decrease patients' willingness to sample via NPS (10). Additionally, specimen collection requires close contact with health care workers, which thereby increases the risk of nosocomial transmission (11, 12). According to recent studies, saliva might be an applicable alternative for the diagnostics of SARS-CoV-2 by RT-PCR in terms of diagnostic accuracy and ease of collection (11–25). Some studies have reported saliva as a more sensitive matrix than NPS for the diagnostic testing of the Omicron variants of SARS-CoV-2 (26–28), whereas others have reported contradictory results (25).

The aim of this study was to validate a laboratory-developed test (LDT) and the PerkinElmer SARS-CoV-2 Plus Kit (no. 3515-0010) for the saliva specimen type. We compared the diagnostic performance of saliva and NPS specimens in the detection of SARS-CoV-2 with RT-PCR in an outpatient setting at the Turku University Hospital drive-in coronavirus testing station by using the LDT and two commercial kits. At the time of specimen collection, Omicron BA.2 was the dominant SARS-CoV-2 variant. To evaluate the tolerability of saliva and NPS sampling, participants were asked to assess both sampling methods with a discomfort scale and to select the preferred sampling method. Additionally, we assessed factors that might influence the sensitivity of saliva sampling.

## RESULTS

We recruited a total of 250 participants. The mean age of participants was 38 (standard deviation [SD], 13; range 12 to 77) years, and 187 (75%) were female. The majority of participants ($N = 246$; 98%) had respiratory or other symptoms, suggesting the possibility of COVID-19. The median duration of symptoms before the collection of specimens was 2 days (interquartile range [IQR], 1 to 3). 18 participants could not produce the saliva specimen volume of 2 mL; 3 of them produced <1 mL of saliva. The median time from specimen collection to nucleic acid extraction was 5 (IQR 3 to 6; range 0 to 16) days. Paired specimens ($n = 9$) that were stored as RNA at 4°C for 3 days gave concordant results in all of the RT-qPCR tests, compared with the rest of the paired specimens ($n = 241$).

With the LDT, SARS-CoV-2 was detected in 135 and 134 participants from NPS and saliva, respectively, with excellent agreement between the specimen types (Cohen's kappa coefficient of 0.911 [95% CI, 0.860 to 0.963]) (Table 1). Of the 135 SARS-CoV-2 positive results with the LDT from NPS, 34 (25%) were Sdel positive, suggesting Omicron BA.1., and 101 (75%) were Sdel negative, suggesting Omicron BA.2. A total of 11 specimens gave discordant results. In six participants, the LDT was positive only from NPS (E-gene $C_T$ values of 36.7, 30.5, 33.2, 34.2, 35.4, and 27.0). In five participants, the LDT was positive only from saliva (E-gene $C_T$ values of 28.0, 27.7, 31.8, 37.5, and 28.6). The sensitivity of the LDT from saliva was 95.6% (95% CI, 90.6% to 98.4%), and the specificity of the LDT from saliva was 95.7% (95% CI, 90.2% to 98.6%).

The LDT E gene $C_T$ values from NPS (median, 20.5; IQR, 18.5 to 24.4) were significantly lower than were the E gene $C_T$ values from saliva (median, 26.2; IQR, 22.3 to 29.5; $P < 0.0001$) (Table 2; Fig. 1A). A moderate correlation was found in the E gene $C_T$ values between NPS and saliva ($r = 0.436$) (Fig. 2A).

**TABLE 1** The SARS-CoV-2 RT-qPCR results obtained from nasopharyngeal swab (NPS) and saliva specimens with a laboratory-developed test, a PerkinElmer SARS-CoV-2 Kit (no. 3501-0010), and a PerkinElmer SARS-CoV-2 Plus Kit (no. 3515-0010)

| Test | NPS | | Total |
|---|---|---|---|
| Laboratory-developed test | | | |
| Saliva | Positive | Negative | Total |
| Positive | 129 | 5 | 134 |
| Negative | 6 | 110 | 116 |
| Total | 135 | 115 | 250 |
| | | | |
| PerkinElmer SARS-CoV-2 | | | |
| Saliva | Positive | Negative | Total |
| Positive | 133 | 3 | 136 |
| Negative | 8 | 106 | 114 |
| Total | 141 | 109 | 250 |
| | | | |
| PerkinElmer SARS-CoV-2 plus | | | |
| Saliva | Positive | Negative | Total |
| Positive | 134 | 2 | 136 |
| Negative | 8 | 106 | 114 |
| Total | 142 | 108 | 250 |

With the PerkinElmer SARS-CoV-2 Kit (3501-0010), SARS-CoV-2 was detected in 141 and 136 participants from NPS and saliva, respectively, with excellent agreement between the specimen types (Cohen's kappa coefficient 0.911 [95% CI, 0.860 to 0.962]) (Table 1). A total of 10 specimens gave discordant results. Similar results were observed with the PerkinElmer SARS-CoV-2 Plus Kit (3515-0010, 11 discordant results) (Table 1). With the PerkinElmer SARS-CoV-2 Kit, the N gene $C_T$ values from NPS (median, 18.7; IQR, 16.9 to 22.6) were significantly lower than the N gene $C_T$ values from saliva (median, 28.3; IQR, 24.1 to 31.6; $P < 0.0001$) (Table 2). With the PerkinElmer SARS-CoV-2 Plus Kit, a similar difference was observed in the N/E gene $C_T$ values between the specimen types (Table 2; Fig. 1B). The ORF1ab $C_T$ values from NPS were significantly lower than the ORF1ab $C_T$ values from saliva with both kits (Table 2; Fig. 1C), and moderate correlation was found between the specimen types (Table 2; Fig. 2C). All of the discordant diagnostic test results from 17 participants as well as the corresponding $C_T$ values are shown in Table S1.

Among the saliva specimens, excellent agreement was observed between the LDT E gene $C_T$ values and the PerkinElmer SARS-CoV-2 Plus Kit N/E gene $C_T$ values ($r = 0.989$) (Fig. 2D) as well as between the LDT E gene $C_T$ values and the PerkinElmer SARS-CoV-2 Plus Kit ORF1ab gene $C_T$ values ($r = 0.987$).

**TABLE 2** Comparison of $C_T$ values between nasopharyngeal swab (NPS) and saliva specimens, with medians and interquartile ranges presented (IQR)

| Method[c] | NPS | | Saliva | | P value[a] | Spearman's correlation coefficient |
|---|---|---|---|---|---|---|
| | Median $C_T$ | IQR | Median $C_T$ | IQR | | |
| LDT RT-qPCR, E gene | 20.5 | 18.5 to 24.4 | 26.2 | 22.3 to 29.5 | <0.0001 | 0.436 |
| LDT RT-qPCR, S gene | 21.9 | 19.8 to 24.1 | 26.3 | 23.0 to 30.0 | 0.022 | 0.375 |
| LDT RT-qPCR, S gene H69-V70 deletion | 28.0 | 24.5 to 31.4 | 30.1 | 27.6 to 36.5 | <0.001 | 0.386 |
| PerkinElmer SARS-CoV-2 kit (no. 3501-0010), N gene | 18.7 | 16.9 to 22.6 | 28.3 | 24.1 to 31.6 | <0.0001 | 0.469 |
| PerkinElmer SARS-CoV-2 kit (no. 3501-0010), ORF1ab gene | 19.3 | 17.6 to 22.8 | 27.0 | 22.9 to 31.0 | <0.0001 | 0.518 |
| PerkinElmer SARS-CoV-2 Plus kit (no. 3515-0010), N and E genes | 17.9 | 16.3 to 22.3 | 26.1 | 22.5 to 30.0 | <0.0001 | 0.519 |
| PerkinElmer SARS-CoV-2 Plus kit (no. 3515-0010), ORF1ab gene | 18.5 | 16.7 to 22.0 | 25.0 | 21.5 to 28.4 | <0.0001 | 0.471 |
| PerkinElmer SARS-CoV-2 Plus kit (no. 3515-0010), S gene H69_V70 deletion | 23.7 | 21.9 to 27.0 | 30.6 | 22.7 to 33.9 | 0.667[b] | 0.500[b] |

[a]Analyzed using the Wilcoxon signed rank test.
[b]Small sample size (NPS, $n = 5$; saliva, $n = 14$). Only three values were correlated. The $P$ value and the correlation coefficient are unreliable.
[c]LDT, laboratory-developed test.

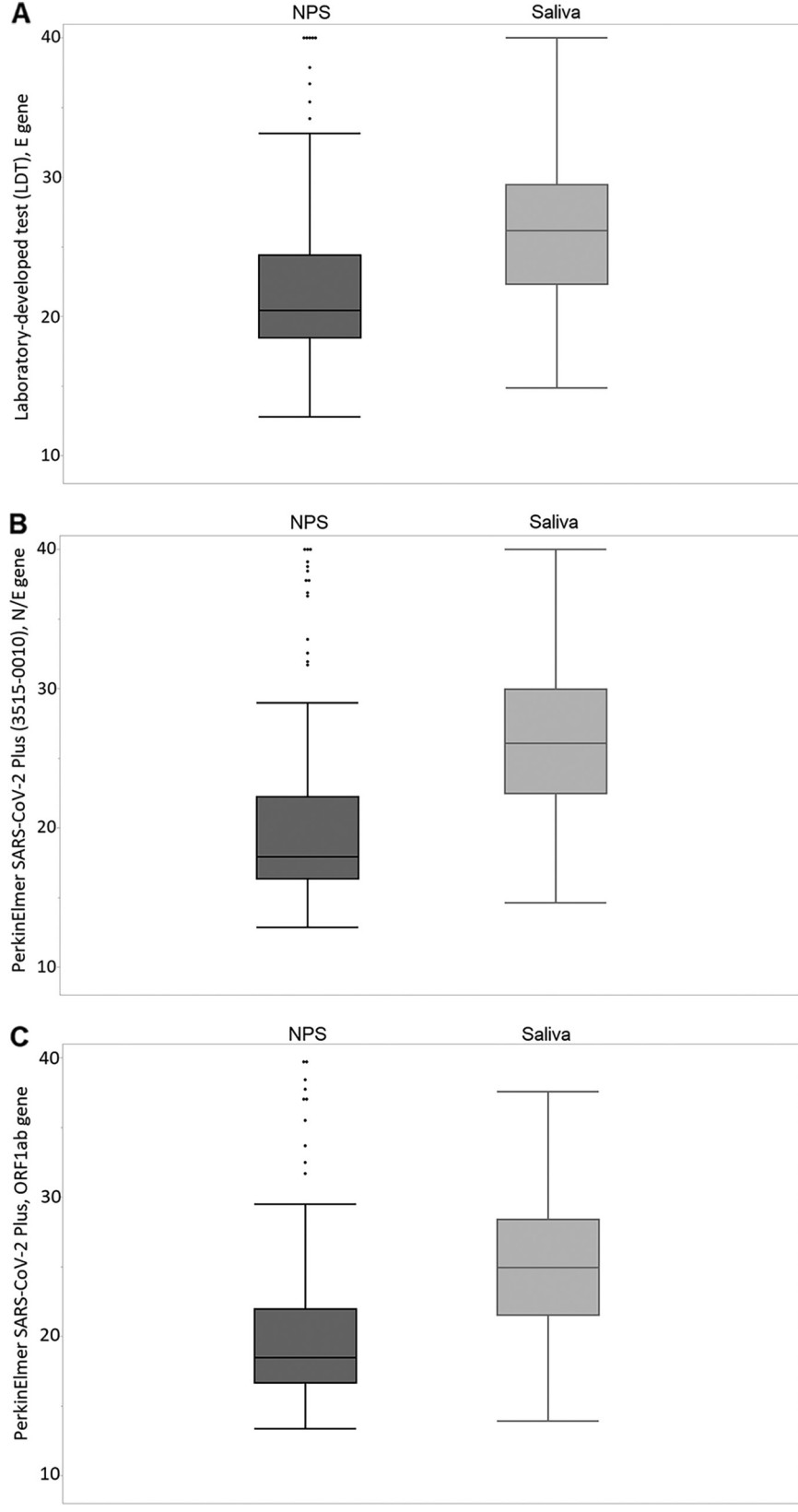

**FIG 1** The SARS-CoV-2 $C_T$ values from nasopharyngeal swab (NPS) and saliva specimens with (A) a laboratory-developed test (LDT), E gene; (B) a PerkinElmer SARS-CoV-2 Plus Kit (no. 3515-0010), N and E genes; or (C) a PerkinElmer SARS-CoV-2 Plus Kit, ORF1ab gene. Medians, interquartile ranges, minimum values, and maximum values are presented.

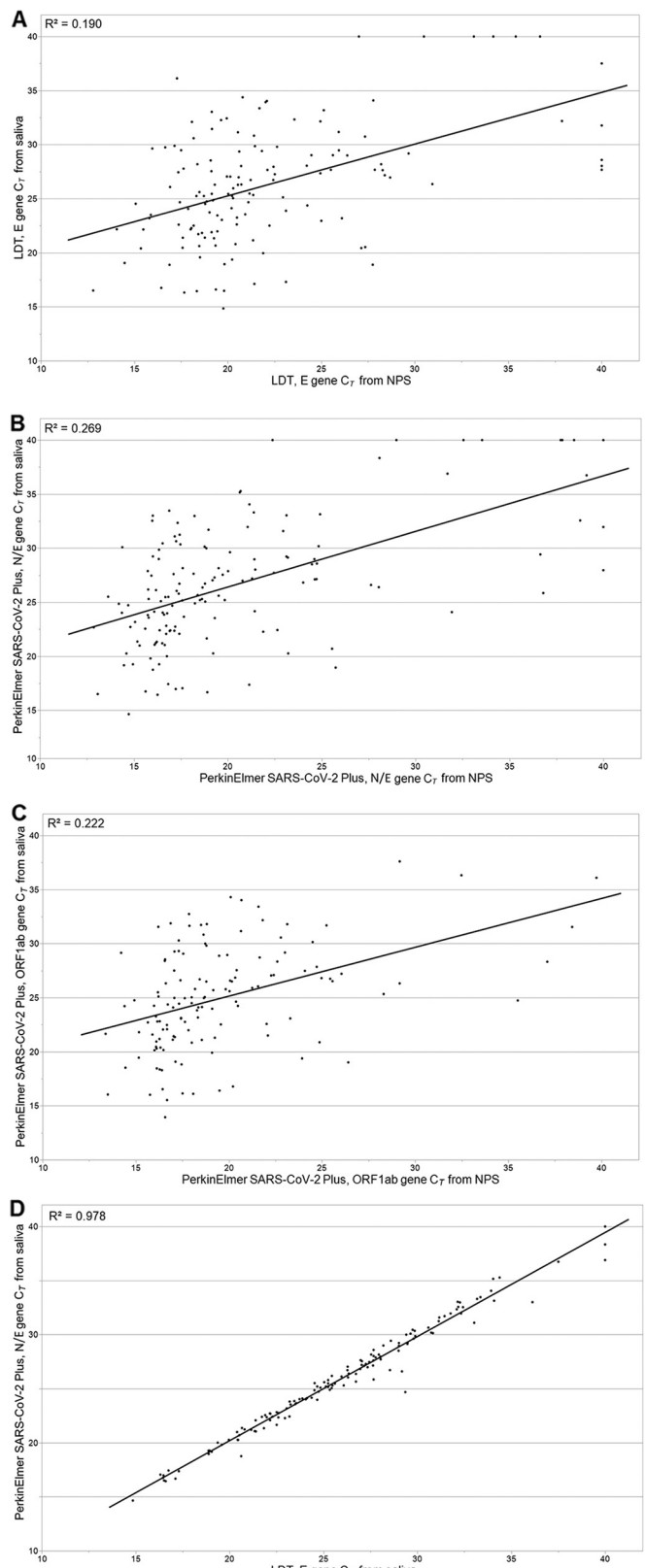

**FIG 2** Correlation of SARS-CoV-2 $C_T$ values between nasopharyngeal swab (NPS) and saliva specimens with (A) a laboratory-developed test (LDT), E gene; (B) a PerkinElmer SARS-CoV-2 Plus Kit (no. 3515-0010), N and E genes; (C) a PerkinElmer SARS-CoV-2 Plus Kit, ORF1ab gene. Panel D presents the correlation between the LDT, E gene, and the PerkinElmer SARS-CoV-2 Plus Kit, N and E gene (same channel), $C_T$ values among saliva specimens.

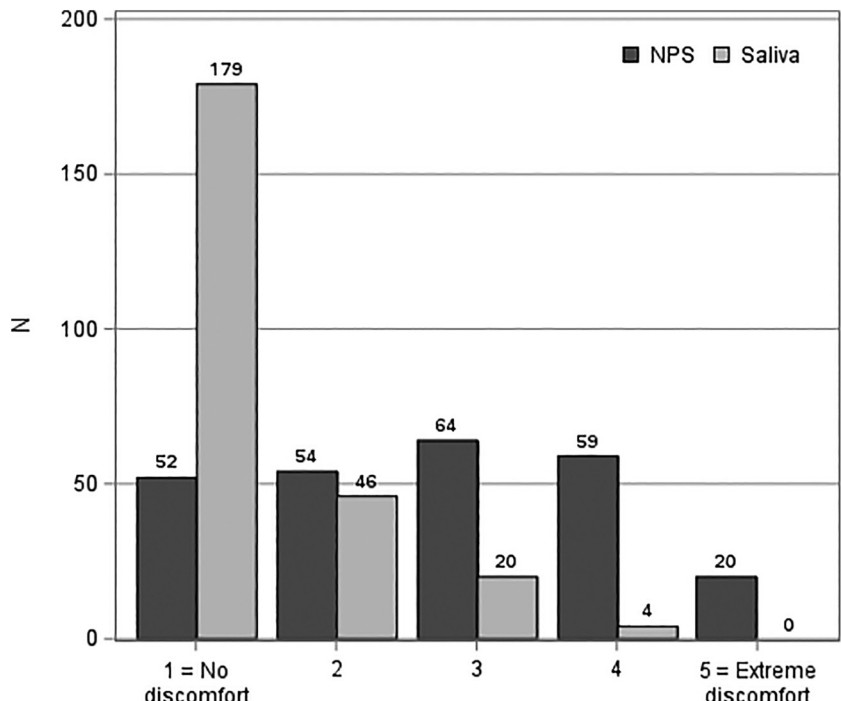

**FIG 3** The discomfort associated with nasopharyngeal (NPS) and saliva sampling. The participants assessed discomfort with a discomfort scale from 1 (no discomfort) to 5 (extreme discomfort). The evaluation is missing from one participant.

Factors that might affect the diagnostic performance of the saliva specimens (drinking, eating, smoking, using mouthwash, and brushing teeth) were evaluated. When any of these had been performed less than 30 min before the saliva sampling, the LDT E gene results were more unreliable, including all 11 of the discordant results ($P < 0.0001$). With the PerkinElmer kit and the Plus kit, 8 and 6 discordant results, respectively, were detected when any of these activities had been performed, compared to 3 and 4 discordant results, respectively, when none of these activities had been performed ($P < 0.0001$). The mean $C_T$ values for participants who had been drinking, eating, smoking, using mouthwash, or brushing teeth 30 min before saliva sampling and participants who had not performed any of these activities were 25.6 and 26.6 ($P = 0.280$) for the LDT E gene, 27.8 and 28.5 ($P = 0.485$) for the PerkinElmer N gene, and 26.4 and 26.7 ($P = 0.797$) for the PerkinElmer Plus N/E genes, respectively.

The majority of the saliva specimens (177/244, 73%; data missing from 6 participants) were collected after 12 a.m. (IQR, 11.00 a.m. to 1.55 p.m.; range, 9.00 a.m. to 6.39 p.m.). The time of day of the saliva specimen collection had no significant effect on the qualitative results of the LDT E gene (odds ratio [OR] = 0.522; 95% CI, 0.161 to 1.693; $P = 0.278$), PerkinElmer kit (OR = 0.687; 95% CI, 0.213 to 2.217; $P = 0.530$), or PerkinElmer Plus kit (OR = 0.789; 95% CI, 0.245 to 2.547; $P = 0.693$). No significant correlations between the specimen collection time and the $C_T$ values of the LDT E gene ($r = 0.072$; $P = 0.400$), PerkinElmer N gene ($r = 0.070$; $P = 0.411$), or PerkinElmer Plus N/E genes ($r = 0.028$; $P = 0.742$) were found.

When evaluating the discomfort associated with NPS and saliva sampling via a discomfort scale from 1 (no discomfort) to 5 (extreme discomfort), a total of 172 (69%) participants gave higher discomfort values to NPS sampling, 17 (7%) gave higher discomfort values to saliva sampling, and 60 (24%) gave equal values for both ($P < 0.0001$) (Fig. 3). A total of 20 participants gave the highest discomfort value for NPS sampling, whereas none gave this value for saliva sampling. When asked which sampling method the participant would choose if further sampling were needed, 148 (59%) participants preferred saliva sampling, 62 (25%) preferred NPS sampling, and 39 (16%) could not decide ($P < 0.0001$).

One participant did not answer the questions about discomfort or the preferred sampling method.

## DISCUSSION

In this prospective study, we collected 250 paired NPS and saliva specimens from voluntary participants in the spring of 2022, when Omicron BA.2 was the dominant SARS-CoV-2 variant in Finland. We compared the diagnostic performance of saliva and NPS specimens in the detection of SARS-CoV-2 by using a LDT with the WHO-recommended primers and probes for the diagnostic gene target (29) as well as two commercial kits. As an important aspect, the tolerability of saliva and NPS sampling was evaluated by asking the participants to assess both sampling methods with a discomfort scale and to select the preferred sampling method. The qualitative results show an excellent sensitivity and specificity of saliva in the detection of SARS-CoV-2 by RT-qPCR, compared with NPS, which is the reference standard for the diagnostic testing of SARS-CoV-2 (3). These findings support the proposition of saliva being an applicable alternative to NPS for the detection of SARS-CoV-2 (30).

Concerning the quantitative results, all of the diagnostic gene targets (E, N, and ORF1ab) consistently showed significantly lower $C_T$ values from NPS than from saliva, indicating that NPS may be slightly more sensitive in the detection of SARS-CoV-2. This result seems to be in line with most previous studies (12, 15, 19, 20, 25, 31, 32). On the other hand, some publications have found that saliva could be more sensitive than NPS for the Omicron BA.1 variant, especially within 5 days of the onset of symptoms (26–28). In our study, the median duration of symptoms before specimen collection was 2 days. Recently, corresponding to our results, Uršič et al. and Salmona et al. reported lower Omicron variant $C_T$ values (i.e., higher viral loads) from NPS than from saliva (25, 32). These specimens were mainly collected in the era of Omicron BA.1, and Uršič et al. concluded that lower $C_T$ values were observed from NPS, regardless of the day of specimen collection (25).

Each of our paired specimens were analyzed with three SARS-CoV-2 tests (LDT RT-qPCR, PerkinElmer SARS-CoV-2 Kit, and PerkinElmer SARS-CoV-2 Plus Kit) for several target genes. The diagnostic target genes E, N, and ORF1ab showed concordant quantitative results from both NPS and saliva. Of interest, among saliva specimens, the $C_T$ values were highly correspondent between the tests, with correlation coefficients exceeding 0.98 when the LDT E gene was compared with the PerkinElmer SARS-CoV-2 Plus Kit N/E genes (measured in the same channel) and the ORF1ab gene, suggesting that the diagnostic performance of the tests from saliva was similar.

Certain factors may influence the diagnostic accuracy of saliva specimens. Some studies suggest that the timing of the saliva specimen collection can affect the viral load, which seems to be the highest in the morning-spit specimen, and thus might reduce the usability of the broadscale utilization of salivary sampling (12, 17, 21, 33). In our study, the time of day of the saliva specimen collection did not affect the qualitative results or the $C_T$ values. In general, when saliva specimens are collected, patients are advised not to eat, drink, or brush teeth before saliva collection to ensure a higher sensitivity of testing (21). However, a study by Callahan et al. suggests that saliva specimens may be equally reliable, even without restrictions to oral hygiene or the time of the sampling (20). Our data show that eating, drinking, smoking, using mouthwash, or brushing teeth 30 min before the saliva collection did not affect the $C_T$ values, which is in concordance with the publication from Callahan et al. However, what we found interesting was that all of the participants with discordant LDT test results from NPS and saliva had been drinking, eating, smoking, using mouthwash, or brushing teeth 30 min prior to specimen collection. This effect was also observed with the PerkinElmer kits. Thus, further research is needed to evaluate the necessity of restrictions prior to saliva collection.

The discomfort associated with NPS sampling has been previously studied (9, 34). However, to our knowledge, patients' burdens of saliva collection have not been previously published, even though saliva sampling is generally considered to be an easier and more tolerable sampling method (11, 12, 14, 18, 19, 31, 35). Unsurprisingly, participants in our study found saliva collection to be significantly less uncomfortable than NPS collection

when assessed with a scale from 1 (no discomfort) to 5 (extreme discomfort). 20 participants assessed NPS collection with a value of 5 (extreme discomfort), compared with none for saliva collection. However, some of the participants had trouble providing the required amount of saliva, which was also observed in a study by Uršič et al. (25). The difficulty to produce enough saliva may have influenced 17 participants to grade saliva sampling as more uncomfortable. Moreover, when the participants selected a preferred sampling method, saliva was chosen by 149 participants, and NPS was chosen by 62 participants. These preferences are in line with the results that were reported by Nacher et al. in their study (31).

The potential advantages of saliva collection, compared with NPS, include less discomfort for most patients as well as easier self-collection (30, 31), which might decrease costs (17). In contrast, the advantages of NPS include faster specimen collection by trained professionals as well as potentially slightly better sensitivity for detecting SARS-CoV-2. Even though saliva seems to be the preferred sampling method in terms of tolerability, a considerable proportion of participants still chose NPS, suggesting that saliva collection is not the easiest method for all. In the future, patients and professionals may be able to choose the most suitable sampling method for each purpose, but further research is required to enable individual specimen collection.

Based on the results of this study, NPS should remain the standard specimen type for the detection of the SARS-CoV-2 Omicron variant, as the lower $C_T$ values of NPS indicate that saliva may be a slightly less sensitive specimen type for SARS-CoV-2 diagnostics. On the other hand, the sensitivity of the saliva specimen in this study was similar to that of NPS, which supports the use of saliva as an alternative to NPS in certain situations. Saliva could reasonably be used for patients who are exposed to complications of NPS collection, are unwilling to sample with NPS, or are particularly sensitive to the discomfort caused by NPS collection. Regarding our observations, saliva could also be collected at any time of day and without strict restrictions to oral hygiene, thereby making the collection at home easy to perform. Sample collection at home might reduce the risk of nosocomial infections, especially in low-income countries, where a lack of protective equipment might be more common.

There are several strengths to our study. The participants were prospectively enrolled in a clinical setting at a drive-in testing station, and paired specimens were collected. The saliva collection was supervised by the study nurse to minimize incorrect collection procedures and to ensure that an adequate volume of saliva was provided. All of the paired specimens were analyzed with three tests, each of which provided consistent results. Additionally, several of the target genes of SARS-CoV-2 were analyzed. We also evaluated the discomfort associated with each sampling method as well as which sampling method each participant preferred.

There are also several limitations to our study. The nucleic acid extraction and RT-qPCRs were not performed on the same day but were performed within a few days of specimen collection. Another limitation is that some saliva specimens did not fulfill the minimum requirement of 2 mL of saliva volume. Additionally, people who experience nasopharyngeal sampling as more painful than others were probably more likely to participate in a study evaluating an alternative sampling method, thereby causing a potential selection bias on the discomfort evaluation. On the other hand, we did not specify that the selection of the preferred sampling method should only be based on the sampling experience. Therefore, some participants might have considered the traditional sampling method more reliable and may thus have preferred NPS, even though it caused more discomfort.

In summary, our study shows the excellent sensitivity and specificity of saliva specimens in the detection of SARS-CoV-2 by RT-qPCR. However, the $C_T$ values of all of the diagnostic target genes were significantly lower from NPS than from saliva, indicating that NPS may be slightly more sensitive in the detection of SARS-CoV-2, compared with saliva. In terms of tolerability, saliva sampling was highly preferred. However, delivering an adequate volume of saliva was challenging for some participants.

**TABLE 3** Primers and probes used in the laboratory-developed test

| Target | Oligonucleotide[a] | Sequence (5′ to 3′)[b] | Position in SARS-CoV-2 reference genome[c] | Concentration in a 25 μL PCR reaction | Reference |
|---|---|---|---|---|---|
| E gene | E_Sarbeco_F1 | ACAGGTACGTTAATAGTTAATAGCGT | 26269-26294 | 400 nM | Corman et al. (29) |
| | E_Sarbeco_R2 | ATATTGCAGCAGTACGCACACA | 26360-26381 | 400 nM | |
| | E_Sarbeco_P1 | FAM-ACACTAGCCATCCTTACTGCGCTTCG-BHQ1 | 26332-26357 | 200 nM | |
| S gene | S_geneF | TCAACTCAGGACTTGTTCTTACC | 21710-21732 | 200 nM | This study |
| | S_geneR | TGGTAGGACAGGGTTATCAAAC | 21796-21817 | 200 nM | |
| | S_geneP | Cy5-CATGCTATACATGTCTCTGGGACCAA-NFQ-02-MGB | 21758-21783 | 80 nM | |
| | S_B117_P | TexasRed-TCCATGCTATCTCTGGGACCAATG-NFQ-02-MGB | 21756-21765, 21772-21785 | 80 nM | |
| Human β-actin | bA-F926 | TTGCCGACAGGATGCAGAA | NA | 40 nM | Mäkelä et al. (36) |
| | bA-R1001 | TCAGGAGGAGCAATGATCTTGAT | NA | 40 nM | |
| | bA-P954 | HEX-TGCCCTGGCACCCAGCACAA-BHQ1 | NA | 80 nM | |

[a]Ordered from Metabion (Metabion International AG, Germany).
[b]FAM: 6-carboxyfluorescein; BHQ: Black Hole Quencher; Cy5: cyanine-5; NFQ: nonfluorescent quencher; MGB: minor groove binder; HEX: hexachlorofluorescein.
[c]NCBI RefSeq: NC_045512.2. NA, not applicable.

## MATERIALS AND METHODS

**Study population and conduct.** We recruited participants at a COVID-19 drive-in testing station of the Turku University Hospital, Turku, Finland. The recruitment period took place between March 9 and April 19 of 2022, when the local COVID-19 epidemic activity was high and Omicron BA.2 was the dominant SARS-CoV-2 variant. According to the Finnish guidelines concerning SARS-CoV-2 testing, the population attending ambulatory SARS-CoV-2 testing mostly consisted of health care workers and patients who were at risk of developing severe COVID-19. The inclusion criteria were: (i) the Participant attends a COVID-19 testing station of the Turku University Hospital for SARS-CoV-2 testing, (ii) the Participant is ≥10 years of age, and (iii) the Participant fluently understands and speaks Finnish (the consent forms for the study were offered only in Finnish). The exclusion criterion was previous participation in this study.

NPS specimens were collected using a Bioer Sample Collection Kit (transport tube with 2 mL of sample preservative fluid; Hangzhou Bioer Technology, Hangzhou, China), according to the instructions of the Turku University Hospital via the attending nurses. Saliva specimens were collected using a DNA/RNA Shield Saliva/Sputum Collection Kit (Zymo Research, USA), according to the kit instructions. In detail, after the NPS had been collected, the study nurse informed the participants of the study, and those who were willing to participate signed written consent. The study nurse asked for the following information: presence of symptoms (if any; symptoms in detail and their duration) and possible eating, drinking, smoking, use of mouthwash, or brushing of teeth during the 30 min prior to the specimen collection. Before saliva sampling, participants were asked to rinse their mouths with water. The study nurse then instructed the participants regarding how to produce a saliva specimen (whole saliva collected via passive drooling). The date and time of the specimen collection were recorded. The collected saliva specimen volume was 2 mL, which was mixed with 2 mL of DNA/RNA shield buffer and vigorously inverted 10 times. If the participant produced less than 2 mL of saliva, DNA/RNA shield buffer was added in the amount 1:1 to the collected saliva specimen volume.

To evaluate the discomfort associated with NPS and saliva sampling, the participants were asked to assess the discomfort caused by each sampling method on a scale from 1 (no discomfort) to 5 (extreme discomfort). After this, the study nurse asked which sampling method the participant would prefer if further COVID-19 testing were needed.

The study protocol was approved by the Ethical Committee of the Hospital District of Southwest Finland (no. 103/1801/2021). All of the study participants provided written informed consent before enrollment. For children <15 years of age, the parent of the child also provided written informed consent.

**Diagnostic methods.** For the nucleic acid extraction and PCR, the specimens were collected in batches of 90 specimens. The NPS specimens were stored at +4°C, and the saliva specimens were stored at room temperature, according to the instructions of the specimen collection kit, for a maximum of 16 days. Prior to extraction, the saliva specimens were heated at 37°C for 10 min. Nucleic acids from 300 μL aliquots of the specimens were extracted using a PerkinElmer Chemagic 360 Nucleic Acid Extractor and a PerkinElmer Chemagic viral DNA/RNA 300 H96 Kit (Wallac Oy, PerkinElmer, Finland), using an elution volume of 90 μL. Extracted RNA of 32 NPS and saliva specimens were stored at −70°C for 6 days due to a strike causing a lack of laboratory personnel to run the specimens. The extracted RNA of 9 NPS and saliva specimens were stored at +4°C for 3 days due to a weekend.

A laboratory-developed SARS-CoV-2 real-time multiplex RT-PCR (RT-qPCR) was performed using the WHO-recommended primers and probe for the diagnostic target, the E gene (29), and the S gene targets to analyze the prevalence of SARS-CoV-2 variants (Table 3). S gene PCR was based on laboratory-designed primers and two competing probes that distinguished whether there was an H69-V70 deletion ($S_{del}$) in the S gene sequence. To measure the human cellular material in the specimens, human β-actin mRNA was amplified (Table 3). The multiplex RT-PCR was performed in 25 μL PCRs, using 9 μL of nucleic acid, Bioline SensiFAST probe one-step master mix (Meridian Bioscience, USA), and a Mic qPCR cycler (Bio Molecular Systems, Australia). The PCR program consisted of a 10 min RT step at 48°C, a 2 min denaturation at 95°C, and 45 cycles at 95°C for 5 s and

58°C for 30 s. The cycle threshold ($C_T$) results from the E gene channel (FAM/green), with threshold set to 0.400, were interpreted as follows: $C_T < 38$, positive; $C_T$ 38 to 39.9, low positive; $C_T \geq 40$, negative. The $C_T$ results from the S gene channel (Cy5/red) and $S_{del}$ channel (TexasRed/orange) were interpreted as follows: $C_T < 42$, positive; $C_T \geq 42$, negative. No fixed threshold levels could be set for the S gene channels due to the various background fluorescence levels. In particular, the $S_{del}$ channel presented a high and variable background fluorescence due to cross-reaction with the S gene without deletion, and, therefore, the $S_{del}$ $C_T$ values are not proportional to the E gene $C_T$ values (Table 2).

The described LDT is the primary diagnostic test for COVID-19 in the Hospital District of Southwest Finland. In routine diagnostics, the E gene is used as the diagnostic target, whereas the S gene results (the S gene with or without the H69-V70 deletion) are used to distinguish the circulating SARS-CoV-2 variants and have been proven to be advantageous in the detection of the appearance of new variants.

Using the nucleic acids from the extractions described above, two commercial SARS-CoV-2 RT-qPCR tests were performed using a Bio-Rad CFX96 Real-Time Cycler, namely, the SARS-CoV-2 RT-qPCR Reagent Kit targeting the N and ORF1ab genes (no. 3501-0010, Wallac Oy, PerkinElmer, Finland) and the SARS-CoV-2 Plus (N/ ORF1ab/E/S 69_70 del) RT-qPCR Reagent Kit targeting the N gene and E gene (measured in the same channel), ORF1ab, and $S_{del}$ for the surveillance of SARS-CoV-2 variants (no. 3515-0010, Wallac Oy, PerkinElmer, Finland).

**Statistical analysis.** The statistical analyses were performed with JMP Pro version 16.2.0 (JMP, United Kingdom). Percentages were compared using a $\chi^2$ test. Means were compared using a (paired) two-sample $t$ test. Medians were compared using a Wilcoxon signed-rank test. Cohen's kappa coefficient was calculated to assess the agreement of the results between the specimen types. For the PCR tests that were performed on the saliva specimens, the sensitivity and specificity (with a 95% confidence interval [CI]) were calculated, using the respective PCR test result from NPS as the reference standard. Pearson's or Spearman's correlation coefficients were used to correlate the $C_T$ values, as applicable. When analyzing the $C_T$ values in cases in which the PCR test was positive from only one specimen type, a $C_T$ value of 40.0 was given for a negative LDT E gene result and for a negative PerkinElmer N and ORF1ab gene result, and a value of 42.0 was given for a negative LDT S gene result and LDT and PerkinElmer $S_{del}$ result. The Cochran-Mantel-Haenszel test was used to evaluate the effect of eating, drinking, smoking, using mouthwash, and brushing teeth on the qualitative results between specimen types. The sample collection time was treated as a continuous variable, and binary logistic regression was used to evaluate the association between the time of day of the saliva specimen collection and the test results from the saliva samples.

## SUPPLEMENTAL MATERIAL

Supplemental material is available online only.
**SUPPLEMENTAL FILE 1**, PDF file, 0.6 MB.

## ACKNOWLEDGMENTS

The Hospital District of Southwest Finland and Wallac Oy, PerkinElmer, made a written study agreement before this study was started. According to the contract, the researchers of the Turku University Hospital were responsible for the interpretation of the results, the drafting of the manuscript, and the submission of the manuscript. The institution (Turku University Hospital) received financial support from Wallac Oy, PerkinElmer, to perform the study. The funder had no role in the data interpretation or in the decision to submit the work for publication.

Kati Mokkala, Siina Vidbäck, and Ville Veikkolainen are PerkinElmer employees.

We have no financial or other relationships that are relevant to the study.

The study nurses, namely, Ulla Torkko and Jenna Torkko, are thanked for recruiting the participants and collecting the discomfort evaluation data from the participants. Tiina Ylinen is thanked for the technical assistance in the laboratory. We thank biostatistician Helena Ollila for the assistance with the statistical analyses and the figures.

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
