## [Reviewer comments · Microbiology Spectrum]

Microbiology Spectrum

Diagnostic Performance and Tolerability of Saliva and Nasopharyngeal Swab Specimens in Detection of SARS-CoV-2 by RT-PCR

Jaakko Ahti, Riikka Österback, Anniina Keskitalo, Kati Mokka, Siina Vidbäck, Ville Veikkolainen, Tytti Vuorinen, Ville Peltola, Antti Hakanen, Matti Waris, and Miia Laine

Corresponding Author(s): Miia Laine, TYKS Turu yliopistollinen keskussairaala

Review Timeline:

Submission Date:	January 4, 2023
Editorial Decision:	February 22, 2023
Revision Received:	March 31, 2023
Accepted:	April 3, 2023

Editor: Paul Luethy

Reviewer(s): Disclosure of reviewer identity is with reference to reviewer comments included in decision letter(s). The following individuals involved in review of your submission have agreed to reveal their identity: Luciana Jesus Costa (Reviewer #1); Priyanka Kapoor (Reviewer #2)

Transaction Report:

DOI: <https://doi.org/10.1128/spectrum.05324-22>

February 22, 2023

Dr. Miia Kristiina Laine
TYKS Turu yliopistollinen keskussairaala
Department of Clinical Microbiology
Kiinamylynkatu 10, PL 52
Turku 20521
Finland

Re: Spectrum05324-22 (Diagnostic Performance and Tolerability of Saliva and Nasopharyngeal Swab Specimens in Detection of SARS-CoV-2 by RT-PCR)

Dear Dr. Miia Kristiina Laine:

Link Not Available

Sincerely,

Paul Luethy

Journals Department
Reviewer comments:

Reviewer #1 (Comments for the Author):

The manuscript by Ahti and co-workers adds another piece of evidence regarding the diagnostic detection of SARS-CoV-2 Omicron (subvariant BA.2) in NPS versus saliva. The authors well demonstrated that although the coefficient of correlation showed good agreement no matter what RT-qPCR kit used (either lab-based or commercial), the Ct ranges when comparing the two types of samples showed a significant higher viral load on NPS, which demonstrates that saliva can be a less sensitive type of sample for SARS-CoV-2 diagnostics, even considering the Omicron circulation. The data also helps to add to the evidence in the field showing that the nasopharyngeal compartment will harbor higher viral loads during SARS-CoV-2 infection,

which was doubted by previous data suggesting higher viral loads for Omicron in the saliva.

The study was based on a reasonable number of specimens (n= 250) and both the study design and generated data are sound. Some points need clarification:

- 1) Authors say that saliva were kept on room temperature for up to 14 days prior to RNA extraction (kit's manufactures' recommendation). It seems quite an extended amount of time to keep saliva containing enveloped viral particles without reducing the amount of viable viral particles. Have authors conducted experiments to assure the viability of these samples? Wouldn't that explain the lower viral load? What was the average time from collection to nucleic acid extraction for samples tested? It was not clear from the sentence that NSP samples were kept at 4C also up to 14 days? Again, seems quite an extensive period to keep viral particles viable.
- 2) Authors mentioned 9 saliva and NPS samples kept for 3 days at 4C after extraction, since RNA is chemically liable and consequently heat degradable, it should be mentioned how these samples performed in RT-qPCR.
- 3) Why is the difference between NPS and saliva on the range of 5-6 Cts higher for E and S targeted genes, but only 2 Cts for the Sdel? is the Median for Sdel in saliva calculated correctly, since the IQR had such a high range? Could authors comment on this data? Also, what was the relevance of using S as target with primers/probe sets to distinguish the del69-70 if this specific wasn't shown?
- 4) Authors should provide a figure or table showing the discordant samples in each RT-qPCR assay they performed, were they the same? Did it vary according to the assay or the sample? How was the internal control performance for discordant samples?
- 5) In lanes 208-211 authors mention that Ct values in saliva were lower for drinking, eating, etc before sample collection, but the difference was not significant, thus they shouldn't say it was different! The sentence should be rephrased to better describe the finding.
- 6) Authors affirm that the time-of-day saliva was collected make no difference for diagnostics, however, no data was shown, specifically, how many periods were evaluated? how were they determined? how many samples foreach period were analyzed? This set of data is of relevance and should be shown.

As a minor observation, the mean age and standard deviation of participants should not be mentioned on the abstract.

Reviewer #2 (Comments for the Author):

reviewer comments in the attached file

Staff Comments:

Preparing Revision Guidelines

Please return the manuscript within 60 days; if you cannot complete the modification within this time period, please contact me. If you do not wish to modify the manuscript and prefer to submit it to another journal, please notify me of your decision immediately so that the manuscript may be formally withdrawn from consideration by Microbiology Spectrum.

Corresponding authors may join or renew ASM membership to obtain discounts on publication fees. Need to upgrade your

membership level? Please contact Customer Service at Service@asmusa.org.

The manuscript by Ahti and co-workers adds another piece of evidence regarding the diagnostic detection of SARS-CoV-2 Omicron (subvariant BA.2) in NPS versus saliva. The authors well demonstrated that although the coefficient of correlation showed good agreement no matter what RT-qPCR kit used (either lab-based or commercial), the Ct ranges when comparing the two types of samples showed a significant higher viral load on NPS, which demonstrates that saliva can be a less sensitive type of sample for SARS-CoV-2 diagnostics, even considering the Omicron circulation. The data also helps to add to the evidence in the field showing that the nasopharyngeal compartment will harbor higher viral loads during SARS-CoV-2 infection, which was doubted by previous data suggesting higher viral loads for Omicron in the saliva.

The study was based on a reasonable number of specimens (n= 250) and both the study design and generated data are sound.

Some points need clarification:

- 1) Authors say that saliva were kept on room temperature for up to 14 days prior to RNA extraction (kit's manufactures' recommendation). It seems quite an extended amount of time to keep saliva containing enveloped viral particles without reducing the amount of viable viral particles. Have authors conducted experiments to assure the viability of these samples? Wouldn't that explain the lower viral load? What was the average time from collection to nucleic acid extraction for samples tested? It was not clear from the sentence that NSP samples were kept at 4C also up to 14 days? Again, seems quite an extensive period to keep viral particles viable.
- 2) Authors mentioned 9 saliva and NPS samples kept for 3 days at 4C after extraction, since RNA is chemically liable and consequently heat degradable, it should be mentioned how these samples performed in RT-qPCR.
- 3) Why is the difference between NPS and saliva on the range of 5-6 Cts higher for E and S targeted genes, but only 2 Cts for the Sdel? is the Median for Sdel in saliva calculated correctly, since the IQR had such a high range? Could authors comment on this data? Also, what was the relevance of using S as target with primers/probe sets to distinguish the del69-70 if this specific wasn't shown?
- 4) Authors should provide a figure or table showing the discordant samples in each RT-qPCR assay they performed, were they the same? Did it vary according to the assay or the sample? How was the internal control performance for discordant samples?
- 5) In lanes 208-211 authors mention that Ct values in saliva were lower for drinking, eating, etc before sample collection, but the difference was not significant, thus they shouldn't say it was different! The sentence should be rephrased to better describe the finding.
- 6) Authors affirm that the time-of-day saliva was collected make no difference for diagnostics, however, no data was shown, specifically, how many periods were evaluated? how were they determined? how many samples foreach period were analyzed? This set of data is of relevance and should be shown.

As a minor observation, the mean age and standard deviation of participants should not be mentioned on the abstract.

1 Diagnostic Performance and Tolerability of Saliva and Nasopharyngeal Swab Specimens in
2 Detection of SARS-CoV-2 by RT-PCR

3

[revised manuscript text omitted]

Materials and Methods

Study population and conduct

We recruited participants at a COVID-19 drive-in testing station of the Turku University
Hospital, Turku, Finland. The recruitment period took place between March 9 and April 19,
2022, when the local COVID-19 epidemic activity was high with Omicron BA.2 as the dominant
SARS-CoV-2 variant. According to the Finnish guidelines concerning SARS-CoV-2 testing, the
population attending ambulatory SARS-CoV-2 testing mostly consisted of healthcare workers
and patients at risk for developing severe COVID-19. The inclusion criteria were: 1) the
Participant attends a COVID-19 testing station of the Turku University Hospital for SARS-CoV-2
testing; 2) the Participant is ≥ 10 years of age; 3) the Participant fluently understands and
speaks Finnish (consent forms for the study offered only in Finnish). The exclusion criterium
was previous participation in this study.

NPS specimens were collected using the Bioer sample collection kit (transport
tube with 2 ml of sample preservative fluid; Hangzhou Bioer Technology, Hangzhou, China)
according to the instructions of the Turku University Hospital by the attending nurses. Saliva
specimens were collected with the DNA/RNA Shield™ Saliva/Sputum Collection Kit (Zymo
Research, USA) according to the kit instructions. In detail, after NPS had been collected, the
study nurse informed participants of the study and those willing to participate signed written

consent. The study nurse asked for the following information: presence of symptoms (if any;
symptoms in detail and their duration); and possible eating, drinking, smoking, using
mouthwash, or brushing teeth during 30 minutes before specimen collection. Before saliva
sampling, participants were asked to rinse their mouths with water. The study nurse then
instructed participants on how to produce a saliva specimen (whole saliva collected by passive
drooling). The date and time of the specimen collection were recorded. The collected saliva
specimen volume was 2 ml, which was mixed with 2 ml of DNA/RNA shield buffer vigorously
inverting 10 times. If the participant could produce less than 2 ml of saliva, DNA/RNA shield
buffer was added in the amount 1:1 to the collected saliva specimen volume.

To evaluate the discomfort associated with NPS and saliva sampling, participants
were asked to assess the discomfort caused by each sampling method on a scale from 1 (no
discomfort) to 5 (extreme discomfort). After this, the study nurse asked which sampling
method the participant would prefer if further COVID-19 testing was needed.

The study protocol was approved by the Ethical Committee of the Hospital District
of Southwest Finland (no. 103/1801/2021). All the study participants provided written informed
consent before enrollment. Regarding children < 15 years of age, the parent of the child also
provided written informed consent.

Diagnostic methods

For nucleic acid extraction and PCR, the specimens were collected in batches of 90 specimens.
NPS specimens were stored at +4 °C and saliva specimens at room temperature according to
instructions of the specimen collection kit for a maximum of 14 days. Prior to extraction, saliva
specimens were heated at 37 °C for 10 min. Nucleic acids from 300 µl aliquots of the specimens

were extracted with PerkinElmer Chemagic™ 360 Nucleic Acid Extractor and PerkinElmer
Chemagic™ viral DNA/RNA 300 H96 kit (Wallac Oy, PerkinElmer, Finland), using an elution
volume of 90 µl. Extracted RNA of 32 NPS and saliva specimens were stored at -70 °C for six
122 days due to a strike causing a lack of laboratory personnel to run the specimens. Extracted RNA
of nine NPS and saliva specimens were stored at +4 °C for three days due to a weekend.

A laboratory-developed SARS-CoV-2 real-time multiplex RT-PCR (RT-qPCR) was
performed using WHO-recommended primers and probe for the diagnostic target, E gene [29],
and S gene targets for analyzing the prevalence of SARS-CoV-2 variants (Table 1). S gene PCR
was based on laboratory-designed primers and two competing probes that distinguished
whether there was an H69-V70 deletion (S_{del}) in the S gene sequence. To measure human
cellular material in the specimens, human β -actin mRNA was amplified (Table 1). The multiplex
RT-PCR was performed in 25 µl PCR reactions, using 9 µl of nucleic acid, Biorad SensiFAST probe
one-step master mix (Meridian Bioscience, USA), and Mic qPCR cycler (Bio Molecular Systems,
Australia). The PCR program consisted of a 10 min RT step at 48°C, a 2 min denaturation at
95°C, and 45 cycles at 95°C for 5 s and 58°C for 30 s. The cycle threshold (C_T) results from the E
gene channel (FAM/green) with threshold set to 0.400 were interpreted as follows: $C_T < 38$
positive; C_T 38 to 39.9 low positive; $C_T \geq 40$ negative. The C_T results from the S gene channel
(Cy5/red) and S_{del} channel (TexasRed/orange) were interpreted as follows: $C_T < 42$ positive; $C_T \geq$
42 negative. No fixed threshold levels could be set for the S gene channels due to varying
background fluorescence levels. Especially the S_{del} channel presented a high and variable
background fluorescence due to cross-reaction with the S gene without deletion, and therefore
the S_{del} C_T values are not proportional to the E gene C_T values (Table 2).

The described LDT is the primary diagnostic test for COVID-19 in the Hospital
District of Southwest Finland. In routine diagnostics, the E gene is used as the diagnostic target,

while S gene results (S gene with or without H69-V70 deletion) are used for distinguishing the
circulating SARS-CoV-2 variants and have been proven advantageous in detecting the
appearance of new variants.

Using the nucleic acids from the extractions described above, two commercial
SARS-CoV-2 RT-qPCR tests were performed with Bio-Rad CFX96 Real-Time Cycler, namely the
SARS-CoV-2 RT-qPCR Reagent kit targeting N and ORF1ab genes (no. 3501-0010, Wallac Oy,
PerkinElmer, Finland) and the SARS-CoV-2 Plus (N/ORF1ab/E/S 69_70 del) RT-qPCR Reagent kit
targeting N gene and E gene (measured in the same channel), ORF1ab, and S_{del} for the
surveillance of SARS-CoV-2 variants (no. 3515-0010, Wallac Oy, PerkinElmer, Finland).

Statistical analysis

The statistical analyses were performed with JMP PRO version 16.2.0 (JMP, UK). Percentages
were compared with the χ^2 test, means with (paired) two-sample t-test, and medians with the
Wilcoxon signed rank test. Cohen's kappa coefficient was calculated to assess the agreement of
results between the specimen types. For the PCR tests performed on saliva specimens,
sensitivity, and specificity (with a 95% confidence interval [CI]) were calculated using the
respective PCR test result from NPS as the reference standard. Pearson's or Spearman's
correlation coefficients were used to correlate C_T values as applicable. When analyzing the C_T
values in cases where the PCR test was positive from only one specimen type, a C_T value of 40.0
was given for a negative LDT E gene result and for a negative PerkinElmer N and ORF1ab gene
result, and a value of 42.0 for a negative LDT S gene result and LDT and PerkinElmer S_{del} result.
The Cochran-Mantel-Haenszel test was used to evaluate the effect of eating, drinking, smoking,
using mouthwash, and brushing teeth on the qualitative results between specimen types.

Binary logistic regression was used to evaluate the association between the time of day of
saliva specimen collection and the test result from saliva.

Results

We recruited a total of 250 participants. The mean age of participants was 38 (standard
deviation [SD], 13; range 12–77) years, and 187 (75%) were female. Majority of participants (N
= 246; 98 %) had respiratory or other symptoms suggesting the possibility of COVID-19. The
median duration of symptoms before collection of specimens was 2 days (interquartile range
[IQR], 1 to 3). Eighteen participants could not produce the saliva specimen volume of 2 ml;
three of them produced saliva < 1 ml.

With the LDT, SARS-CoV-2 was detected in 135 and 134 participants from NPS and
saliva, respectively, with excellent agreement between the specimen types (Cohen's kappa
coefficient 0.911 [95% CI, 0.860 to 0.963]; Table 3a). A total of 11 specimens gave discordant
results. In six participants, the LDT was positive only from NPS (E-gene C_T values, 36.7, 30.5,
33.2, 34.2, 35.4, and 27.0). In five participants, the LDT was positive only from saliva (E-gene C_T
values, 28.0, 27.7, 31.8, 37.5, and 28.6). The sensitivity of the LDT from saliva was 95.6% (95%
CI, 90.6% to 98.4%) and specificity 95.7% (95% CI, 90.2% to 98.6%).

The LDT E gene C_T values from NPS (median 20.5; IQR, 18.5 to 24.4) were
significantly lower than the E gene C_T values from saliva (median 26.2; IQR, 22.3 to 29.5; $P <$
0.0001; Table 2; Fig. 1a). A moderate correlation was found in the E gene C_T values between
NPS and saliva ($r = 0.436$; Fig. 2a).

With the PerkinElmer SARS-CoV-2 kit (3501-0010), SARS-CoV-2 was detected in
141 and 136 participants from NPS and saliva, respectively, with excellent agreement between
the specimen types (Cohen's kappa coefficient 0.911 [95% CI, 0.860 to 0.962]; Table 3b). Similar
results were observed with the PerkinElmer SARS-CoV-2 Plus kit (3515-0010, Table 3c). With
the PerkinElmer SARS-CoV-2 kit, the N gene C_T values from NPS (median 18.7; IQR, 16.9 to 22.6)
were significantly lower than the N gene C_T values from saliva (median 28.3; IQR, 24.1 to 31.6; P
< 0.0001 ; Table 2). With the PerkinElmer SARS-CoV-2 Plus kit, a similar difference was observed
in the N/E gene C_T values between the specimen types (Table 2; Fig. 1b). The ORF1ab C_T values
from NPS were significantly lower than the ORF1ab C_T values from saliva with both kits (Table 2;
Fig. 1c), and moderate correlation was found between the specimen types (Table 2; Fig. 2c).

Among saliva specimens, an excellent agreement was observed between the LDT
E gene C_T values and the PerkinElmer SARS-CoV-2 Plus kit N/E gene C_T values ($r = 0.989$; Fig. 2d),
and between the LDT E gene C_T values and the PerkinElmer SARS-CoV-2 Plus kit ORF1ab gene C_T
values ($r = 0.987$).

Factors that might affect the diagnostic performance of saliva specimens
(drinking, eating, smoking, using mouthwash, and brushing teeth) were evaluated. When any of
these had been performed less than 30 minutes before saliva sampling, the LDT E gene results
were more unreliable, including all the 11 discordant results ($P < 0.0001$). With the PerkinElmer
kit and Plus kit, 8 and 6 discordant results, respectively, were detected when any of these had
been performed compared to 3 and 4 discordant results, respectively, when none of these had
been performed ($P < 0.0001$). The mean C_T values (LDT E gene, 25.6; PerkinElmer N gene, 27.8;
PerkinElmer Plus N/E genes, 26.4) were lower if participants had been drinking, eating,
smoking, using mouthwash, or brushing teeth, compared to the mean C_T values (LDT E gene,

26.6; P = 0.280; PerkinElmer N gene, 28.5; P = 0.485; PerkinElmer Plus N/E genes, 26.7; P =
0.797) when none of these had been performed, but the difference was not significant.

[revised manuscript text omitted]

gene was compared with PerkinElmer SARS-CoV-2 Plus kit N/E genes (measured in the same
channel) and ORF1ab gene, suggesting that the diagnostic performance of the tests from saliva
was similar. should validation of efficacy of LDT be discussed here. Otherwise what is the rationale of using
three kits?

Certain factors may influence the diagnostic accuracy of saliva specimens. Some
studies suggest that the timing of the saliva specimen collection can affect the viral loads, which
seem to be highest in the morning-spit specimen, and thus might reduce the usability of
broadscale utilization of salivary sampling [12, 17, 21, 32]. In our study, the time of day of the
saliva specimen collection did not affect the qualitative results or the C_T -values. In general,
when saliva specimens are collected, patients are advised not to eat, drink, or brush teeth
before saliva collection to ensure higher sensitivity of testing [21]. However, a study by Callahan
et al. suggests that saliva specimens may be equally reliable even without restrictions to oral
hygiene or the time of the sampling [20]. Our data show that eating, drinking, smoking, using
mouthwash, or brushing teeth 30 min before saliva collection did not affect the C_T values,
which is in concurrence with the publication from Callahan et al. However, what we found
interesting was that all the participants with discordant LDT test results from NPS and saliva
had been drinking, eating, smoking, using mouthwash, or brushing teeth 30 minutes prior to
specimen collection. This effect was also observed with the PerkinElmer kits, and thus further
research is needed to evaluate the necessity of restrictions prior to saliva collection.

The discomfort associated with NPS sampling has been previously studied [9, 33].
To our knowledge, however, patients' burden of saliva collection has not been previously
published, even though saliva sampling is generally considered to be easier and a more
tolerable sampling method [11, 12, 14, 18, 19, 30, 34]. Unsurprisingly, participants in our study
found saliva to be significantly less uncomfortable than NPS collection when assessed with a

scale from 1 (no discomfort) to 5 (extreme discomfort). Twenty participants assessed NPS
collection with the value of 5 (extreme discomfort) compared with none for saliva collection.
However, some of the participants had trouble in providing the required amount of saliva,
which was also observed in a study by Uršič et al. [25]. The difficulty to produce enough saliva
may have influenced 17 participants to grade saliva sampling more uncomfortable. Moreover,
when the participants selected the preferred sampling method, saliva was chosen by 149 and
NPS by 62 participants. Preferences are in line with the results reported by Nacher et al. in their
study [30].

The potential advantages of saliva collection compared with NPS include less
discomfort for most patients and easier self-collection [30], which might decrease costs [17]. In
contrast, the advantages of NPS include faster specimen collection by trained professionals,
and potentially slightly better sensitivity for detecting SARS-CoV-2. Even though saliva seems to
be the preferred sampling method in terms of tolerability, a considerable proportion of
participants still chose NPS, suggesting that saliva collection is not the easiest method for all. In
the future, patients and professionals may be able to choose the most suitable sampling
method for each purpose, but further research is required to enable individual specimen
collection.

Please create a separate paragraph of clinical recommendations from this study:
Regarding use of saliva as specimen, timing of sample collection, preference of
patients,

[revised manuscript text omitted]

- 21. Lee Rose A, Herigon Joshua C, Benedetti A, Pollock Nira R, Denkinger Claudia M. 2021.
Performance of Saliva, Oropharyngeal Swabs, and Nasal Swabs for SARS-CoV-2 Molecular
Detection: a Systematic Review and Meta-analysis. *Journal of Clinical Microbiology* 59:e02881-
20.
- 22. European Centre for Disease Prevention and Control. Considerations for the use of saliva
as sample material for COVID-19 testing. 3 May 2021. Stockholm: ECDC; 2021.
- 23. Huang N, Pérez P, Kato T, Mikami Y, Okuda K, Gilmore RC, Conde CD, Gasmi B, Stein S,
Beach M, Pelayo E, Maldonado JO, Lafont BA, Jang S-I, Nasir N, Padilla RJ, Murrah VA, Maile R,
Lovell W, Wallet SM, Bowman NM, Meinig SL, Wolfgang MC, Choudhury SN, Novotny M,

- Aebermann BD, Scheuermann RH, Cannon G, Anderson CW, Lee RE, Marchesan JT, Bush M,
Freire M, Kimple AJ, Herr DL, Rabin J, Grazioli A, Das S, French BN, Pranzatelli T, Chiorini JA,
Kleiner DE, Pittaluga S, Hewitt SM, Burbelo PD, Chertow D, Frank K, Lee J, Boucher RC,
Teichmann SA, et al. 2021. SARS-CoV-2 infection of the oral cavity and saliva. *Nature Medicine*
27:892-903.
- 24. Teo AKJ, Choudhury Y, Tan IB, Cher CY, Chew SH, Wan ZY, Cheng LTE, Oon LLE, Tan MH,
Chan KS, Hsu LY. 2021. Saliva is more sensitive than nasopharyngeal or nasal swabs for
diagnosis of asymptomatic and mild COVID-19 infection. *Scientific Reports* 11: s41598-021-
82787-z.
- 25. Uršič T, Kogoj R, Šikonja J, Roškarič D, Jevšnik Virant M, Bogovič P, Petrovec M. 2022.
Performance of nasopharyngeal swab and saliva in detecting Delta and Omicron SARS-CoV-2
variants. *Journal of Medical Virology* 10:4704-4711.
- 26. Marais G, Hsiao N-Y, Iranzadeh A, Doolabh D, Enoch A, Chu C-Y, Williamson C, Brink A,
Hardie D. 2022. Improved oral detection is a characteristic of Omicron infection and has
implications for clinical sampling and tissue tropism. *Journal of Clinical Virology* 152:105170
- 27. Cornette M, Decaestecker B, Martens GA, Vandecandelaere P, Jonckheere S. 2022. From
Delta to Omicron SARS-CoV-2 variant: Switch to saliva sampling for higher detection rate.
*Journal of Clinical Virology Plus* 2022 2:100090.
- 28. Lai J, German J, Hong F, Tai SHS, McPhaul Kathleen M, Milton Donald K. 2022. Comparison
of Saliva and Midturbinate Swabs for Detection of SARS-CoV-2. *Microbiology Spectrum*
10:e00128-22.
- 29. Corman VM, Landt O, Kaiser M, Molenkamp R, Meijer A, Chu DK, Bleicker T, Brünink S,
Schneider J, Schmidt ML, Mulders DG, Haagmans BL, Van Der Veer B, Van Den Brink S, Wijsman
432 L, Goderski G, Romette J-L, Ellis J, Zambon M, Peiris M, Goossens H, Reusken C, Koopmans MP,
Drosten C. 2020. Detection of 2019 novel coronavirus (2019-nCoV) by real-time RT-PCR.
*Eurosurveillance* 3:2000045.
- 30. Nacher M, Mergeay-Fabre M, Blanchet D, Benois O, Pozl T, Mespouole P, Sainte-Rose V,
Vialette V, Toulet B, Moua A, Saout M, Simon S, Guidarelli M, Galindo M, Biche B, Faurous W,
Chaizemartin L, Fahrasmene A, Rochemont D, Diop F, Niang M, Pujo J, Vignier N, Dotou D,
Vabret A, Demar M. 2021. Diagnostic accuracy and acceptability of molecular diagnosis of
COVID-19 on saliva samples relative to nasopharyngeal swabs in tropical hospital and extra-
hospital contexts: The COVISAL study. *PLOS ONE* 16:e0257169.
- 31. Salmona M, Chaix M-L, Feghoul L, Mahjoub N, Maylin S, Schnepf N, Jacquier H, Walle E-M,
Helary M, Mellon G, Osinski N, Zebiche W, Achili Y, Amarsy R, Mahé V, Le Goff J, Delaugerre C.
2022. Detection of SARS-CoV-2 in Saliva and Nasopharyngeal Swabs According to Viral Variants.
*Microbiology Spectrum* 0:e02133-22.
- 32. Hung DL-L, Li X, Chiu KH-Y, Yip CC-Y, To KK-W, Chan JF-W, Sridhar S, Chung TW-H, Lung K-C,
Liu RW-T, Kwan GS-W, Hung IF-N, Cheng VC-C, Yuen K-Y. 2020. Early-Morning vs Spot Posterior

- Oropharyngeal Saliva for Diagnosis of SARS-CoV-2 Infection: Implication of Timing of Specimen
Collection for Community-Wide Screening. *Open Forum Infectious Diseases* 7:210.
- 33. Williams E, Bond K, Isles N, Chong B, Johnson D, Druce J, Hoang T, Ballard SA, Hall V, Muhi S,
Busing KL, Lim S, Strugnell D, Catton M, Irving LB, Howden BP, Bert E, Williamson DA. 2020.
Pandemic printing: a novel 3D-printed swab for detecting SARS-CoV-2. *Medical Journal of*
*Australia* 213:276-279.
- 34. Kim Y-g, Yun Seung G, Kim Min Y, Park K, Cho Chi H, Yoon Soo Y, Nam Myung H, Lee Chang
454 K, Cho Y-J, Lim Chae S. 2017. Comparison between Saliva and Nasopharyngeal Swab Specimens
for Detection of Respiratory Viruses by Multiplex Reverse Transcription-PCR. *Journal of Clinical*
*Microbiology* 55:226-233.
- 35. Mäkelä M, Öling V, Marttila J, Waris M, Knip M, Simell O, Ilonen J. 2006. Rotavirus-specific T
cell responses and cytokine mRNA expression in children with diabetes-associated
autoantibodies and type 1 diabetes. *Clinical and Experimental Immunology* 145:261-270.

Figure legends

Figure 1. The SARS-CoV-2 C_T values from nasopharyngeal swab (NPS) and saliva specimens with
a) laboratory-developed test (LDT), E gene; b) PerkinElmer SARS-CoV-2 Plus kit (no. 3515-0010),
N and E genes; c) PerkinElmer SARS-CoV-2 Plus kit, ORF1ab gene. Medians, interquartile ranges,
and minimum and maximum values are presented.

Figure 2. Correlation of SARS-CoV-2 C_T values between nasopharyngeal swab (NPS) and saliva
specimens with a) laboratory-developed test (LDT), E gene; b) PerkinElmer SARS-CoV-2 Plus kit
(no. 3515-0010), N and E genes; c) PerkinElmer SARS-CoV-2 Plus kit, ORF1ab gene. Panel d)
presents the correlation between the LDT E gene and PerkinElmer SARS-CoV-2 Plus kit N and E
gene (same channel) C_T values among saliva specimens.

Figure 3. The discomfort associated with nasopharyngeal (NPS) and saliva sampling. The
participants assessed discomfort with a discomfort scale from 1 (no discomfort) to 5 (extreme
discomfort). The evaluation is missing from one participant.

Table 1. Primers and probes used in the laboratory-developed test.

Target	Oligonucleotide ^a	Sequence (5' - 3') ^b	Position in SARS-CoV-2 reference genome ^c	Concentration in a 25 µl PCR reaction	Reference
E gene	E_Sarbeco_F1	ACAGGTACGTTAATAGTTAATAGCGT	26269-26294	400 nM	Corman et al.[29]
	E_Sarbeco_R2	ATATTGCAGCAGTACGCACACA	26360-26381	400 nM	
	E_Sarbeco_P1	FAM- ACACTAGCCATCCTTACTGCGCTTCG- BHQ1	26332-26357	200 nM	
S gene	S_geneF	TCAACTCAGGACTTGTTCTTACC	21710-21732	200 nM	This study
	S_geneR	TGGTAGGACAGGGTTATCAAAC	21796-21817	200 nM	
	S_geneP	Cy5- CATGCTATACATGTCTCTGGGACCAA- NFQ-02-MGB	21758-21783	80 nM	
	S_B117_P	TexasRed- TCCATGCTATCTCTGGGACCAATG-NFQ- 02-MGB	21756-21765, 21772-21785	80 nM	
Human β-actin	bA-F926	TTGCCGACAGGATGCAGAA		40 nM	Mäkelä et al.[35]
	bA-R1001	TCAGGAGGAGCAATGATCTTGAT		40 nM	
	bA-P954	HEX-TGCCCTGGCACCCAGCACAA- BHQ1		80 nM	

477 ^aOrdered from Metabion (Metabion International AG, Germany)

478 ^bFAM: 6-carboxyfluorescein; BHQ: Black Hole Quencher; Cy5: Cyanine-5; NFQ: Nonfluorescent Quencher;

MGB: Minor Groove Binder; HEX: Hexachlorofluorescein

480 ^cNCBI RefSeq: NC_045512.2

Table 2. Comparison of C_T values between nasopharyngeal swab (NPS) and saliva specimens.

Medians are presented with interquartile ranges (IQR).

	NPS		Saliva		P value ^a	Spearman's correlation coefficient
	median C_T	IQR	median C_T	IQR		
LDT RT-qPCR, E gene	20.5	18.5 to 24.4	26.2	22.3 to 29.5	< 0.0001	0.436
LDT RT-qPCR, S gene	21.9	19.8 to 24.1	26.3	23.0 to 30.0	0.022	0.375
LDT RT-qPCR, S gene H69-V70 deletion	28.0	24.5 to 31.4	30.1	27.6 to 36.5	< 0.001	0.386
PerkinElmer SARS-CoV-2 kit (no. 3501-0010), N gene	18.7	16.9 to 22.6	28.3	24.1 to 31.6	< 0.0001	0.469
PerkinElmer SARS-CoV-2 kit (no. 3501-0010), ORF1ab gene	19.3	17.6 to 22.8	27.0	22.9 to 31.0	< 0.0001	0.518
PerkinElmer SARS-CoV-2 Plus kit (no. 3515-0010), N and E genes	17.9	16.3 to 22.3	26.1	22.5 to 30.0	< 0.0001	0.519
PerkinElmer SARS-CoV-2 Plus kit (no. 3515-0010), ORF1ab gene	18.5	16.7 to 22.0	25.0	21.5 to 28.4	< 0.0001	0.471
PerkinElmer SARS-CoV-2 Plus kit (no. 3515-0010), S gene H69_V70 deletion	23.7	21.9 to 27.0	30.6	22.7 to 33.9	0.667 ^b	0.500 ^b

483 ^aAnalyzed with the Wilcoxon signed rank test.

484 ^bSmall sample size (NPS, n = 5; saliva, n = 14). Only three values were correlated. P-value and
 485 correlation coefficient unreliable.

LDT, laboratory-developed test

Table 3. The SARS-CoV-2 RT-qPCR results obtained from nasopharyngeal swab (NPS) and saliva
 specimens with a) laboratory-developed test (LDT); b) PerkinElmer SARS-CoV-2 kit (no. 3501-
 0010); c) PerkinElmer SARS-CoV-2 Plus kit (no. 3515-0010).

a)

LDT	NPS		Total
Saliva	positive	negative	
positive	129	5	134
negative	6	110	116
Total	135	115	250

b)

PerkinElmer SARS-CoV-2	NPS		Total
Saliva	positive	negative	
positive	133	3	136
negative	8	106	114
Total	141	109	250

c)

PerkinElmer SARS-CoV-2 Plus	NPS		Total
Saliva	positive	negative	
positive	134	2	136
negative	8	106	114
Total	142	108	250

Figure 1. The SARS-CoV-2 C_T values from nasopharyngeal swab (NPS) and saliva specimens with a)
laboratory-developed test (LDT), E gene; b) PerkinElmer SARS-CoV-2 Plus kit (no. 3515-0010), N
and E genes; c) PerkinElmer SARS-CoV-2 Plus kit, ORF1ab gene. Medians, interquartile ranges, and
minimum and maximum values are presented.

Figure 2. Correlation of SARS-CoV-2 C_T values between nasopharyngeal swab (NPS) and saliva
specimens with a) laboratory-developed test (LDT), E gene; b) PerkinElmer SARS-CoV-2 Plus kit
(no. 3515-0010), N and E genes; c) PerkinElmer SARS-CoV-2 Plus kit, ORF1ab gene. Panel d)
presents the correlation between the LDT E gene and PerkinElmer SARS-CoV-2 Plus kit N and E
gene (same channel) C_T values among saliva specimens.

Figure 3. The discomfort associated with nasopharyngeal (NPS) and saliva sampling. The
participants assessed discomfort with a discomfort scale from 1 (no discomfort) to 5 (extreme
discomfort). The evaluation is missing from one participant.

March 31st, 2023

Spectrum05324-22 (Diagnostic Performance and Tolerability of Saliva and Nasopharyngeal Swab Specimens in Detection of SARS-CoV-2 by RT-PCR)

Response to Reviewers:

Reviewer #1

The manuscript by Ahti and co-workers adds another piece of evidence regarding the diagnostic detection of SARS-CoV-2 Omicron (subvariant BA.2) in NPS versus saliva. The authors well demonstrated that although the coefficient of correlation showed good agreement no matter what RT-qPCR kit used (either lab-based or commercial), the Ct ranges when comparing the two types of samples showed a significative higher viral load on NPS, which demonstrates that saliva can be a less sensitive type of sample for SARS-CoV-2 diagnostics, even considering the Omicron circulation. The data also helps to add to the evidence in the field showing that the nasopharyngeal compartment will harbor higher viral loads during SARS-CoV-2 infection, which was doubted by previous data suggesting higher viral loads for Omicron in the saliva.

The study was based on a reasonable number of specimens (n= 250) and both the study design and generated data are sound.

- We thank the Reviewer for their appraisal. We did our best in preparing point-by-point responses (found below) to the raised issues. We hope that these responses fulfill Your expectations.

Some points need clarification:

1) Authors say that saliva were kept on room temperature for up to 14 days prior to RNA extraction (kit's manufactures' recommendation). It seems quite an extended amount of time to keep saliva containing enveloped viral particles without reducing the amount of viable viral particles. Have authors conducted experiments to assure the viability of these samples? Wouldn't that explain the lower viral load? What was the average time from collection to nucleic acid extraction for samples tested? It was not clear from the sentence that NSP samples were kept at 4C also up to 14 days? Again, seems quite an extensive period to keep viral particles viable.

- We thank the Reviewer for this important comment. The time from collection to nucleic acid extraction for the paired saliva and NPS samples of each patient were: 0 days for 12 paired samples, 2 days for 13 paired samples, 3 days for 47 paired samples, 4 days for 42 paired samples, 5 days for 56 paired samples, 6 days for 53 paired samples, 7 days for 18 paired samples, and 16 days for 9 paired samples. The extended amount of time before extraction of 9 (3.6%) paired samples was caused by the shortage of laboratory personnel during the COVID-19 pandemic. The median time from collection to nucleic acid extraction was 5 (IQR 3-6) days. We have added the following sentence to the Results section (lines 177-178 of the Marked-Up Manuscript): "*The median time from specimen collection to nucleic acid extraction was 5 (IQR 3-6; range 0-16) days.*"

According to the CE-IVD validated DNA/RNA Shield™ Saliva/Sputum Collection Kit manufacturer's protocol saliva can be stored at room temperature for up to 28 days. The maximum time from sample collection to RNA extraction was 16 days instead of 14 days due to a weekend but still well within the manufacturer specified 28 days. We apologize the error and we have corrected the following sentence in the Diagnostic methods paragraph, line 118: "*NPS specimens were stored at +4 °C and saliva specimens at room temperature according to instructions of the specimen collection kit for a maximum of 16 days.*"

Considering that the saliva samples were stored at the validated specimen collection kit and the paired saliva and NPS samples had the same storage time before extraction, it would seem unlikely that the lower viral load of saliva samples was caused by the storage time. In addition, Ott et al. have studied the stability of SARS-CoV-2 RNA at different temperatures and found that SARS-CoV-2 RNA detection remained stable for up to 25 days even from nonsupplemented saliva (Ott IM, Strine MS, Watkins AE, et al, Stability of SARS-CoV-2 RNA in Nonsupplemented Saliva, Emerg Infect Dis. 2021;27(4):1146–1150. doi: 10.3201/eid2704.204199).

2) Authors mentioned 9 saliva and NPS samples kept for 3 days at 4C after extraction, since RNA is chemically liable and consequently heat degradable, it should be mentioned how these samples performed in RT-qPCR.

- We thank the Reviewer for the comment. Of these 9 samples, concordant qualitative test results were obtained from both saliva and NPS (4 samples were positive with all three tests from both specimen types). No difference was observed in the LDT E gene Ct values from saliva between these 9 samples (median 29.3; IQR, 22.7 to 32.6) and the rest 241 samples (median 26.0; IQR, 22.3 to 29.4; P=0.49). Similarly, no difference was observed in the LDT E gene Ct values from NPS between the 9 samples (median 23.2; IQR, 19.7 to 25.7) and the rest of the samples (median 20.4; IQR, 18.5 to 24.2; P=0.79).

With the PerkinElmer SARS-CoV-2 kit (3501-0010), the N gene Ct values from saliva did not differ between the mentioned 9 samples (median 31.5; IQR, 24.4 to 33.1) and the rest of the samples (median 28.2; IQR, 24.2 to 31.5; P=0.62). Similar results were found from NPS (the 9 samples; median 20.5; IQR, 18.2 to 23.0; and the rest of the samples; median 18.6; IQR, 16.9 to 22.6; P=0.93).

With the PerkinElmer SARS-CoV-2 Plus kit (3515-0010), no difference was observed in the ORF1ab Ct values from saliva between the 9 samples (median 28.2; IQR, 22.1 to 31.4) and the rest of the samples (median 24.9; IQR, 21.5 to 28.3; P=0.43). Similar results were found from NPS (the 9 samples; median 19.9; IQR, 18.1 to 22.4; and the rest of the samples; median 18.4; IQR, 16.6 to 22.0; P=0.88).

We have added the following sentence to the Results section, lines 178-180: "*Paired specimens (n=9) that were stored as RNA at +4 °C for three days gave concordant results in all RT-qPCR tests when compared with the rest of the paired specimens (n=241).*"

3) Why is the difference between NPS and saliva on the range of 5-6 Cts higher for E and S targeted genes, but only 2 Cts for the Sdel? is the Median for Sdel in saliva calculated correctly, since the

IQR had such a high range? Could authors comment on this data? Also, what was the relevance of using S as target with primers/probe sets to distinguish the del69-70 if this specific wasn't shown?

- We thank the Reviewer for lifting this up. In the LDT, E gene served as the diagnostic target. The qualitative analysis of two S gene targets (S gene and Sdel) was used for epidemiologic surveillance of new SARS-CoV-2 variants in Southwest Finland. As we have described in the Materials and Methods section (lines 138-141), varying background due to cross-reactivity of S gene with the Sdel probe was observed in the Sdel channel (TexasRed/orange). The threshold level was manually set above the background fluorescence and the threshold was at a higher level compared to the other gene targets, resulting in the higher C_T values for the Sdel analysis as compared to other gene targets. We checked the calculation and found no errors and no obvious explanation for the small C_T difference between NPS and saliva for Sdel. The number of SARS-CoV-2 variants with Sdel was smaller (n=34; suggesting Omicron BA.1) and the spread of their C_T values larger as compared to variants without the deletion (n=101; suggesting Omicron BA.2), increasing the weight of individual values. We have added these numbers of variants to the Results section, lines 183-185: *“Of the 135 SARS-CoV-2 positive results with the LDT from NPS, 34 (25%) were Sdel positive, suggesting Omicron BA.1. and 101 (75%) Sdel negative, suggesting Omicron BA.2.”*

4) Authors should provide a figure or table showing the discordant samples in each RT-qPCR assay they performed, were they the same? Did it vary according to the assay or the sample? How was the internal control performance for discordant samples?

- We thank the Reviewer for raising this important issue. A total of 17 participants provided discordant samples. The LDT and PerkinElmer SARS-CoV-2 kit both gave 11 discordant results while the PerkinElmer SARS-CoV-2 Plus kit gave 10 discordant results. Many of the discordant results occurred with the same participants: Six (35%) were discordant between all kits (4 NPS pos/saliva neg; 2 saliva pos/NPS neg); three (18%) were discordant between two kits; and eight (47%) were discordant between only one kit. Of the samples with discordant results, seven samples gave only C_T values ≥ 35 .

As a response, we have changed the manuscript as follows: We have added a Supplementary Appendix that includes a complete table of discordant samples and their C_T values. In the Results section, we have added the total number of participants, which provided discordant results (lines 204-206). We have also added the number of discordant results obtained with the PerkinElmer SARS-CoV-2 kit and PerkinElmer SARS-CoV-2 Plus kit (lines 196-198).

5) In lanes 208-211 authors mention that C_t values in saliva were lower for drinking, eating, etc before sample collection, but the difference was not significant, thus they shouldn't say it was different! The sentence should be rephrased to better describe the finding.

- We thank the Reviewer for lifting this up. We have corrected the sentence as follows: *“The mean C_T values for participants who had been drinking, eating, smoking, using mouthwash, or brushing teeth 30 minutes before saliva sampling and participants who had not performed any of these activities were 25.6 and 26.6 ($P=0.280$) for the LDT E gene, 27.8 and 28.5 ($P=0.485$) for the PerkinElmer N gene, and 26.4 and 26.7 ($P=0.797$) for the PerkinElmer Plus N/E genes, respectively.”*

6) Authors affirm that the time-of-day saliva was collected make no difference for diagnostics, however, no data was shown, specifically, how many periods were evaluated? how were they determined? how many samples for each period were analyzed? This set of data is of relevance and should be shown.

- We thank the Reviewer for this valuable comment since we were unclear in explaining how we interpreted the association between the sample collection time and diagnostic sensitivity. First, the time of the sample collection was recorded with one-minute precision. In the statistical analysis, binary logistic regression was used to evaluate the association between the time of day of saliva sample collection and the qualitative test result or C_T values from saliva. In these calculations, time is a continuous variable, which is compared to the qualitative test results of the saliva samples (categorical variable) and to the C_T values from saliva (continuous variable). That is to say, the time of sample collection was not divided into different periods (e.g., morning or afternoon) and statistical calculations were performed only using time as a continuous variable.

We have now also inspected our results using time of collection as a categorical variable. The time of collection was divided into two periods: before 12.00 (morning; number of collected samples, 67) and from 12.00 onward (afternoon and evening; number of collected samples, 177); data missing from 6 participants. The time of collection ranged from 9.00 am to 6.39 pm. Fisher's exact test (X^2 test) was used to compare categorical time and qualitative results of the LDT E gene ($P = 0.39$), PerkinElmer N gene ($P = 0.56$), and PerkinElmer Plus N/E genes ($P = 0.56$). Two sample t-test was used to compare categorical time and C_T values from saliva. No significant difference was seen for the mean C_T values of the LDT E gene ($P = 0.20$), PerkinElmer N gene ($P = 0.22$), and PerkinElmer Plus N/E genes ($P = 0.32$). We have clarified the use of time as a continuous variable to the Statistical analysis paragraph of the Materials and Methods section (lines 166-167): "*The sample collection time was treated as a continuous variable, and binary logistic regression was used to evaluate the association between the time of day of saliva specimen collection and the test result from saliva.*"

We have also added the range of sample collection time, and the number of collected samples per period (morning vs afternoon and evening) to the Results section (lines 227-228): "*The majority of saliva specimens (177/244, 73%; data missing from six participants) were collected after 12 am (IQR, 11.00 am to 1.55 pm; range, 9.00 am to 6.39 pm).*" We have not added the categorical calculations of time to the findings to our revised manuscript, as the evaluation of the time of saliva sample collection was not the main focus of this manuscript and we think that our calculation using time as a continuous variable would describe the finding well enough.

As a minor observation, the mean age and standard deviation of participants should not be mentioned on the abstract.

- We thank the Reviewer for the observation. We have removed the mean age and standard deviation of participants from the abstract.

Reviewer #2

1) Why was LDT chosen as one of the ways to test target genes? Was validation of LDT with two commercial kits one of the objectives? IF yes, it is not mentioned in the objectives of the study?

- We thank the Reviewer for the comment. In the very beginning of the pandemic in February-March 2020, the described LDT was developed in the Hospital District of Southwest Finland because no commercial SARS-CoV-2 PCR tests were available. So far the LDT has been the primary diagnostic test for COVID-19 in Southwest Finland. The LDT from nasopharyngeal specimen has been validated and accredited by the Finnish Accreditation Service (testing laboratory no. T148; according to the standard SFS EN ISO 15189:2013).

The aim of this study was to validate the LDT and the PerkinElmer SARS-CoV-2 Plus kit (no. 3515-0010) for saliva sample type. The PerkinElmer SARS-CoV-2 Plus kit has been previously validated for NPS. The PerkinElmer SARS-CoV-2 kit (3501-0010) has been validated for both saliva and NPS and served as the reference test for the PerkinElmer SARS-CoV-2 Plus kit. Regarding the LDT, the test results obtained from NPS served as the reference for the test results obtained from saliva.

We have added the following sentence to the Introduction section (lines 69-70 of the Marked-Up Manuscript): *“The aim of this study was to validate a laboratory-developed test (LDT) and the PerkinElmer SARS-CoV-2 Plus kit (no. 3515-0010) for saliva specimen type.”*

2) Missing relevant reference: Kapoor P, Chowdhry A, Kharbanda OP, Popli DB, Gautam K, Saini V. Exploring salivary diagnostics in COVID-19: a scoping review and research suggestions. *BDJ Open* 2021; 7: 8. <https://doi.org/10.1038/s41405-021-00064-7>

- We thank the Reviewer for the observation. We have added the recommended reference to the Discussion section (lines 256 and 305) and to the References of the manuscript (lines 463-464).

3) Should validation of efficacy of LDT be discussed here. Otherwise what is the rationale of using three kits?

- We thank the Reviewer for lifting this up. As mentioned in our response to the Comment no. 1, the aim of this study was to validate the LDT and the PerkinElmer SARS-CoV-2 Plus kit (no. 3515-0010) for saliva specimen type. This objective has been added to the Introduction section. The rationale of using three kits was to evaluate the performance of different diagnostic target genes (E, N, and ORF1ab) from saliva.

Regarding the validation of commercial kits, the reference kit should be CE-IVD marked (Medical Device Coordination Group guidance; https://health.ec.europa.eu/system/files/2022-02/mdcg_2021-21_en.pdf). In addition, when alternative specimen types, e.g. saliva in this study, are validated for the detection of SARS-CoV-2, their performance should be compared to test results obtained from NPS. Due to these requirements, the SARS-CoV-2 RT-qPCR Reagent Kit (no. 3501-0010; validated for both NPS and saliva) served as the reference test for the SARS-CoV-2 Plus kit (no. 3515-0010) test results from saliva. As we have discussed in the manuscript, the different diagnostic target genes showed concordant qualitative and quantitative test results from saliva and from NPS, suggesting that the diagnostic performance of the LDT and the two commercial kits was similar.

4) Please create a separate paragraph of clinical recommendations from this study: Regarding use of saliva as specimen, timing of sample collection, preference of patients.

- We thank the Reviewer for this valuable remark since we were insufficient in explaining the generalizability of this research. As a response, we have added the following paragraph to the Discussion section of the manuscript (lines 313-323): *“Based on the results of this study, NPS should remain to be the standard specimen type for the detection of the SARS-CoV-2 Omicron variant since the lower C_T values of NPS indicate that saliva may be a slightly less sensitive specimen type for SARS-CoV-2 diagnostics. On the other hand, the sensitivity of the saliva specimen in this study was similar to NPS, which supports the use of saliva as an alternative to NPS in certain situations. Saliva could reasonably be used for patients that are exposed to complications of NPS collection, are unwilling to sample with NPS, or are particularly sensitive to the discomfort caused by NPS collection. Regarding our observations, saliva could also be collected at any time of day and without strict restrictions to oral hygiene, thus making the collection at home easy to perform. Sample collection at home might reduce risk of nosocomial infections, especially in low-income countries where a lack of protective equipment might be more common.”*

April 3, 2023

Dr. Miia Kristiina Laine
TYKS Turu yliopistollinen keskussairaala
Department of Clinical Microbiology
Kiinamylynkatu 10, PL 52
Turku 20521
Finland

Re: Spectrum05324-22R1 (Diagnostic Performance and Tolerability of Saliva and Nasopharyngeal Swab Specimens in Detection of SARS-CoV-2 by RT-PCR)

Dear Dr. Miia Kristiina Laine:

Your manuscript has been accepted, and I am forwarding it to the ASM Journals Department for publication. You will be notified when your proofs are ready to be viewed.

Sincerely,

Paul Luethy
Editor, Microbiology Spectrum